# RNA-Seq Analysis of Protection against Chronic Alcohol Liver Injury by *Rosa roxburghii* Fruit Juice (Cili) in Mice

**DOI:** 10.3390/nu14091974

**Published:** 2022-05-09

**Authors:** Shan Yang, Xian-Yu Huang, Nian Zhou, Qin Wu, Jie Liu, Jing-Shan Shi

**Affiliations:** Kay Laboratory of Basic Pharmacology and Joint International Research of Ethnomedicine of Ministry of Education, Zunyi Medical University, Zunyi 563000, China; yangshanshiyan@163.com (S.Y.); xianyuhuang11@163.com (X.-Y.H.); 15585702194@163.com (N.Z.); wuqinzmc@163.com (Q.W.)

**Keywords:** *Rosa roxburghii* Tratt. fruit juice (Cili), chronic alcohol liver injury, triglyceride, Oil-red O staining, RNA-Seq, qPCR

## Abstract

*Rosa roxburghii* Tratt. fruit juice (Cili) is used as a medicinal and edible resource in China due to its antioxidant and hypolipidemic potentials. The efficacy of Cili in protecting alcohol-induced liver injury and its underlying mechanism was investigated. C57BL/6J mice received a Lieber-DeCarli liquid diet containing alcohol to produce liver injury. After the mice were adapted gradually to 5% alcohol, Cili (4 mL and 8 mL/kg/day for 4 weeks) were gavaged for treatment. The serum enzyme activities, triglyceride levels, histopathology and Oil-red O staining were examined. The RNA-Seq and qPCR analyses were performed to determine the protection mechanisms. Cili decreased serum and liver triglyceride levels in mice receiving alcohol. Hepatocyte degeneration and steatosis were improved by Cili. The RNA-Seq analyses showed Cili brought the alcohol-induced aberrant gene pattern towards normal. The qPCR analysis verified that over-activation of CAR and PXR (Cyp2a4, Cyp2b10 and Abcc4) was attenuated by Cili. Cili alleviated overexpression of oxidative stress responsive genes (Hmox1, Gsta1, Gstm3, Nqo1, Gclc, Vldlr, and Cdkn1a), and rescued alcohol-downregulated metabolism genes (Angptl8, Slc10a2, Ces3b, Serpina12, C6, and Selenbp2). Overall, Cili was effective against chronic alcohol liver injury, and the mechanisms were associated with decreased oxidative stress, improved lipid metabolism through modulating nuclear receptor CAR-, PXR-and Nrf2-mediated pathways.

## 1. Introduction

*Rosa roxburghii* Tratt. fruit is known as “Cili” in China. It grows well in the mountainous region of Guizhou, Southwest China and has been used for medicinal remedies and healthy foods since ancient times. The effect of Cili in promoting digestion was firstly recorded in “Ben-Cao-Gang-Mu-Shi-Yi” in 1765 A.D. The decoction of the Ci-Li-Gen is recorded in “Zhonghua Bencao” for chronic gastritis, stomach ache, acute enteritis, diarrhoea, and white diarrhoea in human and livestock [1]. Cili is listed in Pharmacopeia of China as a medicinal herb [2]. In recent years, Cili juice (also called prickly pear juice) is used as nutritional beverage and exerts antioxidant and antihyperlipidemic potentials as functional foods. Phytochemical analysis has revealed that Cili is a good source of essential nutrients (e.g., sugars, proteins, vitamins, inorganic salts, and various essential amino acids), and contains polysaccharides, polyphenols especially rich in flavonoid, triterpenoids, and organic acids [3,4,5]. It is especially rich in vitamin C, superoxide dismutase and phenolic compounds. At least 52 compounds contained in Cili have been identified [4]. Shan-Wang-Guo prickly pear juice (Cili) uses organic *Rosa roxburghii* Tratt. fresh fruit as a raw material, undergoes eight organic production processes, and is subjected to the high frequency sterilization process. It retains the nutritional composition and natural fragrance of *Rosa roxburghii* Tratt. fruit without adding sugar and preservatives (http://www.shanwangguo.cn/ accessed on 20 January2022).

*Rosa roxburghii* Tratt. fruit has been shown to have numerous beneficial effects on health, including antioxidant, anti-ageing, anti-atherosclerotic, antitumor, antihyperlipidemia, hypoglycemia in diabetes, and modulation of gut microbiota as recently reviewed by Wang et al. [4]. Liver is the main organ of the body detoxification and metabolism system and liver lesions are common clinical diseases. Juice of *Rosa roxburghii* Tratt. fruit has been demonstrated to alleviate high-fat-diet induced hyperlipidemia in mice [6,7]; a hydro-alcoholic extract from *Rosa roxburghii* Tratt. fruit decreases hyperlipidemia in high-fat-fed rats [8]. *Rosa roxburghii* Tratt. fruit is identified as a promising dietary fibre source, which produces butyrate and affects microbiota composition [9]. Polysaccharide from *Rosa roxburghii* Tratt. fruit attenuates hyperglycemia and hyperlipidemia in diabetic db/db mice [10], and *Rosa roxburghii* Tratt. fruit juice ameliorates arsenic-induced liver damage in rats [11]. However, little is known about Cili on chronic alcohol-induced liver injury.

Alcohol binge drinking is the seventh leading risk factor for both death and societal burden of alcoholism [12,13]. Alcoholic liver injury is the main alcohol-related disease, from simple steatosis to alcoholic steatohepatitis (inflammation), progressive liver fibrosis, cirrhosis, and liver cancer [13]. Alcohol is mainly oxidized in the liver and produces a large number of oxidative metabolites such as acetaldehyde and acetic acid that cause oxidative damage to proteins, lipids, and nucleic acids. Cytochrome P450 2E1 (Cyp2e1), located in the microsomes, plays a major role in the metabolism of alcohol to generate reactive oxygen species that contribute to liver injury and in turn induce downstream Cyp2a5 and the Nrf2 pathways [14]. The Lieber-DeCarli liquid diet containing alcohol is one of the most widely used experimental models to study chronic alcohol liver diseases in rodents. It is an easy, accurate, reliable, and inexpensive model to study the pathogenesis of alcohol liver diseases in experimental settings [15].

In this study, the Lieber-DeCarli liquid diet containing 5% alcohol was used to induce chronic alcoholic injury. Cili from Shan-Wang-Guo prickly pear juice Co., (Guizhou, China) was given to mice at 4 mL/kg and 8 mL/kg for 4 weeks to determine the protective effects of Cili against chronic alcohol liver injury. In addition to routine measures of liver injury such as serum enzyme activities, blood and liver triglyceride levels, liver histopathology and Oil-red O staining for steatosis, RNA-Seq analysis was performed to explore molecular mechanisms, and qPCR analysis was used to verify selected gene expressions. The results clearly demonstrated the protective effects of Cili against chronic alcohol damage and provided a molecular mechanism for the protection.

## 2. Materials and Methods

### 2.1. Cili, Liquid Diets and Chemicals

The *Rosa roxburghii* Tratt. fruit juice (prickly pear juice, Cili) was from Shan-Wang-Gou Co., (Guizhou, China). The Lieber-DeCarli liquid diet was from TROPHIC Animal Feed High-Tech Co., Ltd. (Haian, China); the 95% analysis pure ethanol was from Tianjin Kemiou Chemical Reagent Co., Ltd. (Tianjin, China); the isopropanol, anhydrous ethanol, and chloroform were from Sichuan East Chemical Group, China; the biochemistry kits were from Nanjing Jiancheng Bioengineering Institute (Nanjing, China).

### 2.2. Experimental Animals

Six-week-old male C57BL/6J mice (20–22 g) were purchased from Zhejiang Vital River Experimental Animal Technology Co., Ltd. (Hangzhou, China). Mice were housed in an accredited animal facility at the Key Lab of Basic Pharmacology, Zunyi Medical University. The mice had free access to food and water, with room temperature of 21–23 °C, humidity at 48–52%, and lighting for 12 h–12 h alternately. All experiments were carried out in accordance with Chinese guidelines for animal welfare and were approved by the Animal Use and Care Committee of Zunyi Medical University (SYXK2021-0003).

After one week of adaptation, mice were randomly divided into Control (10), Model (12), M+Cili-L (12), and M+Cili-H (12) groups. Control mice were given the liquid control diet without alcohol, and the remaining groups were adapted to the Lieber-DeCarli liquid diet with gradual increase in alcohol from 1% on the first day to 5% on the fifth day and thereafter. On the 6th day, Cili 4 mL/kg (M+Cili-L) and 8 mL/kg (M+Cili-H) was gavaged daily for four consecutive weeks. Control and Model group mice were given 10 mL/kg of saline. The dose of Cili was based on the literature [6,7,11] and on data from our preliminary studies against alcoholic liver injury. Animal body weights and general health conditions were monitored every two to three days.

Four hours after the last dose of Cili administration, mice were euthanized, and blood was collected via orbital venous plexus bleeding. The livers were isolated, weighed, and stored in −80 °C for RNA analysis.

### 2.3. Serum Enzyme Activities

The blood was stood for 2 h and centrifuged at 3000 rpm for 10 min to isolate serum. The serum activities of alanine transference (ALT) and aspartate transferase (AST) were determined with a Multiskan Go full wavelength microplate reader (Thermo Fisher, Durham, NC, USA) according to the ALT and AST kit instructions (Jiancheng, Nanjing, China).

### 2.4. Triglyceride Determination

The livers were homogenized with 0.9% saline to prepare 10% homogenates. The liver homogenates were centrifuged at 3000 rpm for 10 min to collect supernatant. The triglyceride content in the serum and liver supernatant were determined with a triglyceride kit from Nanjing Jiancheng Bioengineering Institute, following the instructions of the manufacturer.

### 2.5. Haematoxylin and Eosin Staining and Oil-Red O Staining

A piece of the liver was fixed in 10% buffered formalin for 48 h at room temperature, embedded in paraffin at 60 °C, and sectioned into 3.5-µm-thick sections using a RM2235 microtome (Leica Microsystems GmbH, Weitzlar, Germany). The sections were deparaffinized in xylene and rehydrated using a gradient of ethanol (100, 95, 85 and 75%). The histological sections were then stained with haematoxylin and eosin at room temperature for 8–10 min and 4–5 s, respectively. The images of slices were observed with an Olympus light microscope (Olympus Corporation, Tokyo, Japan).

For fatty liver detection, liver tissue was embedded in OCT and stored at −80 °C. The OCT-embedded samples were sectioned at 4 μm and stained with Oil-Red O for the evaluation of fat droplets under the Olympus light microscope.

### 2.6. RNA Extraction and Easy RNA-Seq

Total RNA was extracted from the liver using Trizol reagent (Takara, Takara Islands, Japan) according to the manufacturer’s instructions. The quantity and quality of RNA were determined with a NanoDrop 2000 Ultra micro-spectrometer (Thermo Fisher, Durham, NC, USA). The RNA samples were reverse transcribed with Oligo dT primer to produce cDNA. The generated first-strand cDNA was co-reacted by the RNase H enzyme, DNA polymerase and T4 ligase to generate double-stranded cDNA; the double-stranded cDNA was fragmented by the Tn5 enzyme and add the remedial design (RD) sequence required for sequencing at both ends. The sequencing primers at both ends of P5 and P7 were connected by RD sequences and enrichment PCR amplification was performed. Successful library construction was sequenced. Finally, bioinformatic analyses of differentially expressed genes from the RNA-Seq data between groups were performed by Weilang Biotechnology Co., Ltd. (Chongqing, China).

### 2.7. Principal Component Analysis

Principal Component Analysis (PCA) was performed to visualize the gene expression patterns. The total number of RNA sequencing genes (22,500/sample) was imported into Partek Flow Server (Partek Inc., St. Louis, MO, USA) with Control, Model, M+Cili-L, and M+Cili-H groups (4 samples/group). Images of the PCA were generated to visualize the gene distribution patterns.

### 2.8. Heatmap Visualization of Differentially Expressed Genes

The differentially expressed genes (DEGs) were analysed via the DESeq2 method by Weilang Biotechnology (Chongqing, China) under the criteria of FDR (Padj) < 0.05 as compared to the Control group. The V-Lookup was used to align DEGs generated from Model vs Control, M+Cili-L vs Control, and M+Cili-H vs Control groups, and the comparison file was then imported into TreeView version 1.6 (https://treeview.software.informer.com/1.6/ accessed on 20 January2022) to generate heatmap for visualization.

### 2.9. Real-Time qPCR

Total RNA was transcribed using the PrimeScriptTM RT reagent Kit (Takara, Japan). Real-time qPCR was performed on the CFX 96 Real-time fluorescence Quantitative PCR instrument (Bio-Rad, Inc., Hercules, CA, USA) using SYBR Green dye for relative quantification of gene expression. The mouse primers were designed by online Primer3 and synthesized by Sangon Bioengineering Co., Ltd. (Shanghai, China) (Appendix A). The relative gene expression level was calculated by the 2^−ΔΔCT^ method, and mouse Gapdh was used as the housekeeping gene. The specificity of each gene expression was confirmed by the melting curve with a single peak, and the stability of the Gapdh of 32 samples from four groups had a mean ± SD of 19.79 ± 0.63.

### 2.10. Statistics

Differentially expressed genes (DEG) from the RNA-Seq analysis were analysed by the DESeq2 method and compared with the Control group, with a *p* value < 0.05 considered significant. The serum enzyme activities, triglyceride levels, and qPCR data were expressed as Mean ± SEM. One-way analysis of variance (One-way ANOVA) was used to determine statistical differences between groups, followed by Dunn’s or Tukey multiple range tests. The significance level was set at *p* < 0.05.

## 3. Results

### 3.1. Cili Protected against Alcohol-Induced Elevation of Triglyceride

After mice were adapted to 5% alcohol in a Lieber-DeCarli liquid diet, Cili was gavaged from the 6th day to the 34th day for 28 days (4 weeks). Figure 1A shows the body weight changes during the experiment. Compared to the Control group, the body weight gain of mice fed a Lieber-DeCarli liquid diet containing 5% alcohol was decreased. No apparent difference in body weights between the Model and Cili groups was evident except for a slight difference of the M+Cili-H group at 17 and 20 days of the experiment. At the end of the experiments, the body weights were 25.16 ± 0.40, 20.21 ± 0.45, 20.76 ± 0.58 and 19.21 ± 0.59 g for the Control, Model, M+Cili-L, and M+Cili-H groups, respectively (Figure 1A), the liver weights were 1074 ± 18, 1002 ± 31, 878 ± 38 and 904 ± 34 mg for the Control, Model, M+Cili-L, and M+Cili-H groups, respectively, resulting in an increased liver index (Liver/body weight, mg/g) by alcohol feeding (Figure 1B), and Cili at the low dose prevented the liver index increase by alcohol.

Alcohol feeding increased serum ALT (Figure 2A) and AST (Figure 2B) activities. Cili at both doses tended to decrease serum ALT, but was not significant, while it did not affect the AST activity compared to the Model group.

Alcohol feeding increased serum triglyceride content (Figure 3A) and tended to increase liver triglyceride (Figure 3B). Cili at both doses decreased serum triglyceride compared to the Model group; Cili at the high dose also decreased liver triglyceride (Figure 3B).

### 3.2. Cili Improved Alcohol-Induced Pathology and Lipid Accumulation

The liver sections stained with haematoxylin and eosin (Figure 4, top) showed that the Model group had fat vacuoles, liver degeneration and foci of apoptosis/necrosis compared with the Control group; these pathologic lesions were greatly improved in both Cili groups. Oil-Red O staining (Figure 4, bottom) revealed extensive accumulation of lipid droplets in the Model group; which was markedly reduced by Cili. The effects of low and high doses of Cili on lipid droplet accumulation were similar (Figure 4).

### 3.3. Cili Reversed Alcohol-Induced Aberrant Gene Expression

The RNA-Seq analysis generated 22500 gene targets. The Principle Component Analysis (PCA) is shown on Figure 5A. The PCA value is 69.39 %, with PC1 = 40.94%, PC2 = 12.05%, and PC3 = 10.40%. The distributions of the Control, M+Cili-L, and M+Cili-H groups were apparently separated from the Model group, while the Control and M+Cili-L groups had largely the same pattern. The differentially expressed genes (DEGs) were analysed with the DESeq2 method and compared to the Control group. The DEGs were screened using FDR < 0.05. The Model group had 804 up-regulated genes, 407 down-regulated genes, the M+Cili-L group had 78 up-regulated genes and 79 down-regulated genes, while the M+Cili-H group had 1023 up-regulated genes and 388 down-regulated genes (Figure 5B).

Based on 1211 DEGs from the Model group compared to the Control group, the M+Cili-L group had 127 GEGs and the M+Cili-H group had 283 DEGs compared to the Control group (Appendix A). Heatmap comparisons among groups are shown in Figure 6, where red represents upregulated genes and blue represents downregulated genes. Cili intervention apparently attenuated or reversed alcohol-induced aberrant gene expressions.

### 3.4. The qPCR Analysis of Selected Genes

Based on the fold change and alcohol liver injury associated with the nuclear receptor biomarkers, Cyp2b10, Cyp2a4, and Abcc4 (typical CAR and PXR biomarkers) were further analysed using real-time qPCR (Figure 7). The expression of Cyp2b10 was increased 2100-fold by alcohol, but attenuated with low dose of Cili to 116-fold and high dose of Cili to 10-fold. The expression of Cyp2a4 was increased 50-fold by alcohol, but attenuated in the M+Cili-L group to 13-fold and in the M+Cili-H group to 6-fold. The expression of Abcc4 was increased 48-fold by alcohol, but decreased to 11-fold with M+Cili-L, and 3.5-fold with M+Cili-H. A clear Cili dose-dependent effect was evident.

Based on common oxidative biomarkers in alcohol liver injury, the expressions of Hmox1, Gstm3, Gsta1, Cdkn1a, Vidlr, Nqo1, Gclc, and Mt1 were further analysed using real-time qPCR as shown in Figure 8. The expression of Hmox1 was increased 20-fold by alcohol but was attenuated with Cili-L and Cili-H to 2–3-fold. The expression of Gstm3 was increased 17-fold by alcohol but was attenuated with Cili-L to 4.4-fold and Cili-H to 2.9-fold. The expression of Gsta1 was increased 61-fold by alcohol but decreased to 14-fold with a low dose of Cili, and 3.3-fold with a high dose of Cili. The expression of Cdkn1a was increased 6-fold by alcohol but decreased to 4-fold with a high dose of Cili and prevented by a low dose of Cili. The expression of Vldlr was increased 5-fold by alcohol but decreased to 2-fold with two doses of Cili. The expression of Nqo1 was increased 11-fold by alcohol but decreased to 5-fold with two doses of Cili. The expression of Gclc was increased 3-fold by alcohol but decreased to 1.6-fold with a low dose of Cili and prevented with a high dose of Cili. The expression of Mt1 was increased 2–3-fold among all three groups.

Based on the common lipoprotein and lipid metabolism biomarkers, the expressions of Cyp4a14, Acta2, Angptl8, Slc10a2, Ces3b, Serpina12, C6, and Selenbp2 were further analysed using real-time qPCR as shown in Figure 9. The expression of Cyp4a14 was increased 14.5-fold, a low dose of Cili had no effect, while a high dose of Cili decreased it to 8.8-fold. The expression of Acta2 was increased 3-fold by alcohol, but the increase was prevented by both doses of Cili. The expression of the key lipid metabolism gene Angptl8 was decreased to 25%, and both doses of Cili brought it to 92% and 105%, respectively. The expression of Slc10a2 was decreased to 24% of the Control group, and Cili-L and Cili-H brought it to 48% and 70% of the Control group, respectively. The expression of Ces3b was decreased to 23% of the Control group by alcohol, and it was recovered to 42 and 49% by low and high doses of Cili, respectively. The expression of Serpina12 was decreased to 8% by alcohol and was recovered by the high dose of Cili to 19%. The expression of C6 was decreased to 11% by alcohol but recovered by both doses of Cili to 20%. The expression of Selenbp2 was decreased to 17%, and the high dose of Cili brought it to 27%.

## 4. Discussion

The present study clearly demonstrated the protective effects of Cili against chronic alcohol liver injury, as evidenced by serum and liver triglyceride levels, and importantly by haematoxylin and eosin and Oil-red O staining. The RNA-Seq analysis revealed an aberrant gene expression pattern produced by chronic alcohol which was alleviated by Cili towards that of the Control group. The qPCR of selected genes verified the RNA-Seq results, the over-activated CAR and PXR was attenuated by Cili. The oxidative stress genes induced by chronic alcohol were ameliorated by Cili and lipid metabolism genes suppressed by alcohol were recovered to various degrees by Cili. To our knowledge, this is the first research to demonstrate beneficial effects of Cili against chronic alcohol liver injury, and the altered gene expressions could provide molecular targets against chronic alcohol diseases.

### 4.1. Cili Protected against Chronic Alcohol Liver Injury

The general health and biochemical changes from chronic exposure of C57BL/6 mice to the Lieber-DeCarli liquid diet containing alcohol are characterized by increased liver index, mild increase in serum enzyme activities, and elevations in triglycerides in serum and liver [15,16,17]. The present study replicated all these changes indicating the successful establishment of a chronic alcohol liver injury model. Although both doses of Cili did not alleviate alcohol-induced body weight loss, the low dose Cili decreased the liver index. Cili at both doses tended to decrease the elevated ALT and AST, although not statistically significant. On the other hand, Cili at both doses prevented the increase in serum glycerides, and Cili at the high dose decreased liver triglycerides, even below that of the Control group. Generally speaking, Cili at both doses was effective in ameliorating the alcohol-increased liver index, ALT, and triglyceride, but no dose-response was evident.

Liver degeneration and steatosis are the main pathological features of chronic alcohol liver injury [16,17,18], characterized by incomplete cell structure, disordered cell arrangement, blurred boundaries and large space, hepatocyte vacuolation, and hepatocyte fatty degeneration. Cili at both doses greatly improved liver pathological lesions. The Oil-red O staining is a common method to determine lipid accumulation in alcohol-induced fatty liver in rodents [18]. Extensive lipid droplets were evident in the Model mouse liver, and Cili at both doses prevented lipid accumulation in the liver (Figure 4), indicating the protection against alcohol fatty liver.

### 4.2. Cili Alleviated Chronic Alcohol-Induced Aberrant Gene Expression

Principal component analysis (PCA) is a technique for exploratory data analysis and for making predictive models [19]. The PCA of 16 RNA-Seq samples revealed distinct gene expression patterns: the alcohol Model group was separated from the Control group, Cili at low dose brought the gene expression towards the Control group, while Cili at high dose distributed it away from the Control group, but opposite to the Model group (Figure 5A). Compared to the Control group and based on *p*-value <0.05, chronic alcohol produced 2716 DEGs, while a low dose of Cili had 619 DEGs, and high dose of Cili had 911 DEGs (Appendix A). It can be concluded that the gene expression difference between the Control and the Model groups was large, while both Cili groups had less DEGs, and attenuated the magnitude of increased or decreased genes as visualized in the heatmap (Figure 6).

Real-time qPCR was performed to verify the RNA-Seq results. The first category was the constitutive androstane receptor (CAR) and pregnane X receptor (PXR) biomarkers Cyp2b10, Cyp2a4, and Abcc4 (Figure 7). Alcohol dramatically increased the expression of the marker genes over 50+ fold, which were markedly attenuated by both groups of Cili. The CAR can be activated by dietary flavonoids and participates in glucose and lipid metabolism [20]. The CAR^−/−^ mice showed increased sensitivity to chronic alcohol-induced liver injury, however, over-activation of CAR by the CAR agonist TCPOBOP enhanced hepatotoxicity in both acute and chronic alcohol exposures [21]. The PXR-null mice are resistant to chronic alcohol-induced hepatosteatosis and gene expression changes in a clear PXR-dependent manner [22]. The functions of CAR and PXR are overlapped in xenobiotic responses including chronic alcohol [23]. The Cyp2b10, Cyp2a4, and Abcc4 biomarkers are all CAR and PXR target genes [23,24]. Polyphenols (resveratrol and ellagic acid) protect against chronic alcohol-induced fatty liver in Wild-type but not in CAR-null mice, implying that appropriate CAR activation is beneficial in protecting against alcohol liver injury [25]. It should be pointed out that Cyp2a5 was similarly increased as Cyp2a4 in this study. Alcohol-induced Cyp2e1 co-localized with Cyp2a5 and preceded the induction of Cyp2a5. The Cyp2a5 knockout mice exhibited an enhanced alcoholic liver injury and hyperglycemia compared to the Wild-type control mice, suggesting the protective effects of Cyp2a5 against alcohol-induced oxidative liver injury [14]. Thus, activation of CAR and PXR could also be considered as an adaptive response to alcohol exposure within the threshold range.

The second category is the oxidative stress responsible genes (Figure 8). These genes include Nrf2 antioxidant pathway genes, glutathione detoxification genes, and non-enzymatic antioxidant gene metallothionein. For example, the Miao ethnomedicine *Penthorum chinense* Pursh protected against Lieber-DeCarli alcohol diet induced liver injury by upregulating Nrf2 and its downstream antioxidant protein Ho-1, and by downregulating Cyp2e1 [26]. Similarly, Curcumin or Shangxi aged vinegar attenuates chronic alcohol liver injury via induction of the Nrf2 pathway gene Nqo1, Ho-1, Sod, Gpx through ERK/p38/Nrf2 antioxidant signalling pathways to suppress inflammatory responses [27,28]. The activation of Nrf2 also induces the downstream gene Vldlr in the liver and adipose tissue to protect against alcohol-induced oxidative stress and hepatocyte injury [29]. Ursolic acid, a triterpenoid and a strong Nrf2 activator, protected against chronic alcohol liver injury by inducing Nqo1, Gclc and glutathione S-transferases [30]. Glutathione detoxification enzymes including Gsta1, Gstm3, Gstm1, Gstp1, Gclm, and Gclc play important roles in protecting against alcoholic liver damage and diseases [31,32]. Broccoli sprout extract alleviates alcohol-induced oxidative liver injury through activation of Nrf2 and glutathione detoxification system [33]. Cdkn1a encodes P21 and is increased during chronic alcohol feeding and contributes to modulate alcoholic fatty liver in mice via the ROS-HNE-P21 pathway [34]. Metallothionein is a cysteine-rich, low-molecular weight protein shown to protect against alcohol liver injury [35]. Similar to the activation of CAR and PXR, activation of Nrf2 and glutathione detoxification system could also be considered as adaptive responses to alcohol exposure within the threshold range, but their over-activation could lead to liver injury. Both doses of Cili attenuated their over-expression, although no clear dose-response, implying a reduced oxidative damage to the liver.

The third category is the lipid metabolism genes (Figure 9). Cytochrome P450 omega-hydroxylase 4a14 (Cyp4a14) plays an important role in the lipid metabolism and its overexpression is implicated in pathogenesis of alcoholic fatty liver disease. In the present study, Cyp4a14 was induced 14-fold, which was attenuated to 8.8-fold with the high dose of Cili. The Cyp4A antagonist HET0016 attenuated Cyp4a10 and Cyp4a14 induction by alcohol and prevented alcohol-induced fatty liver [36]. Smooth muscle α actin (Acta2) induction is associated with liver injury and transforms resident hepatic stellate cells into liver specific myofibroblasts for cellular motility and contraction [37]. In the present study, alcohol increased Acta2 3-fold, which was prevented by both doses of Cili. Angiopoietin-like protein 8 (Angptl8) is a novel important regulator in metabolic disorders, and plays a crucial role in lipid metabolism [38]. Chronic alcohol decreased the expression of Angptl8 by 75%, and Cili rescued such a downregulation. The apical sodium-dependent bile acid transporter (ASBT, Slc10a2) is important in the enterohepatic cycling of bile acids and plays an important role in fatty liver diseases [39]. Chronic alcohol also decreased its expression to 24% of the Control group, and Cili-L and Cili-H doses brought its expression towards normal (48% and 70% of the Control group, respectively). Carboxylesterase 3B (Ces3b) hydrolyses long-chain fatty acids and thioesters that would play a role in the lipid metabolism. The Ces3b helps provide substrates for the assembly of very low-density lipoprotein (VLDL) in the liver. Downregulation of Ces3b by high-fat diet in obese mice could be reversed by the fermentation extract of the larva of the edible insect *Tenebrio molitor* with amelioration of fatty liver [40]. Similarly, Cili administration recovered alcohol-induced downregulation of Ces3b to certain extent. Serpin family A member 12 (Serpina12, also called vaspin) is protective against alcohol-induced steatosis but is decreased in serum and liver tissues in alcoholic patients and animals [41]. A chronic alcohol diet markedly decreased its expression, and the high dose of Cili rescued it from 8% to 19% of the Control group. Complement C6-deficient rats were more sensitive to alcohol-induced liver damage that C6+/+ rats [42]. In the present study, C6 was decreased 90% by chronic alcohol, and was slightly recovered by both doses of Cili (from 11% to 20% of the Control group). The selenium binding protein 2 (Selenbp2) and Selenbp1 are involved in lipid metabolism via the peroxisome proliferator-activated receptor-α pathway [43]. Cili at the high dose slightly ameliorated the downregulation of Selenbp2. Overall, Cili was beneficial in recovery of these lipid metabolism and steatosis-related genes in alcohol-induced liver damage.

### 4.3. Multiple Components in Cili Could Function in an Intergraded Manner to Achieve Beneficial Effects

The comprehensive review on Cili in Food and Function [4] clearly demonstrated that Cili is not only a good source of essential nutrients, but also contains polysaccharides, polyphenols, triterpenoids, and organic acids and at least 52 beneficial compounds. Various raw extracts from *Rosa roxburghii* Tratt. fruit [8,44,45], including polysaccharide [46,47], polyphenols [48,49], triterpenoids [30] and many other components [3,4,5] have been found to exert beneficial effects. Juice of *Rosa roxburghii* Tratt. fruit (Cili) has been proven to be very effective in alleviating high-fat-diet induced hyperlipidemia in mice [6,7] and arsenic-induced liver damage in rats [11]. It is very likely that multiple components in Cili could exert their biological functions in an integrated way to target multiple targets to achieve beneficial effects of Cili beverage as functional foods.

## 5. Conclusions

The present study is among the first to demonstrate the protective effects of Cili against chronic alcohol liver injury by routine measures, especially by serum and liver triglyceride levels, histopathology, and Oil-red O staining. The RNA-Seq and qPCR analyses further revealed the underlining mechanisms, including the attenuation of over-activated CAR, PXR, and Nrf2 pathways, and amelioration of metabolism disturbance. This study provided experimental evidence for Cili as functional foods to protect against alcohol binge drinking and improve metabolic disorders through multiple molecular events (Figure 10) to achieve liver protection.

## Figures and Tables

**Figure 1 nutrients-14-01974-f001:**
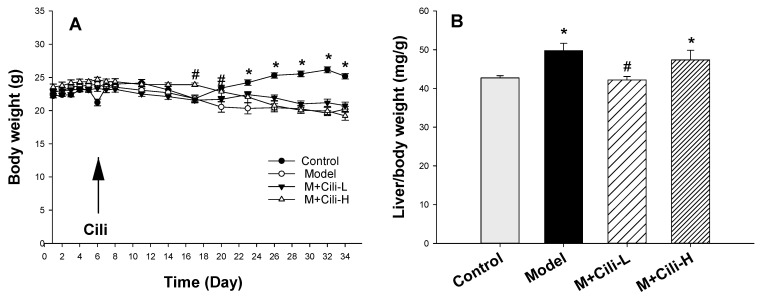
Animal body weight (**A**) and liver index (**B**). Mice were adapted to gradient alcohol increase from 1% to 5% in a Lieber-DeCarli liquid diet from Day 1 to Day 5, and Cili was gavaged at the 6th day (arrow) at 4 mL/kg (M+Cili-L) and 8 mL/kg (M+Cili-H) daily for 28 days (4 weeks). Livers were collected at the end of the experiment. Data are mean ± SEM (*n* = 9), * Significantly different from the Control group, *p* < 0.05; # Significantly different from the Model group, *p* < 0.05.

**Figure 2 nutrients-14-01974-f002:**
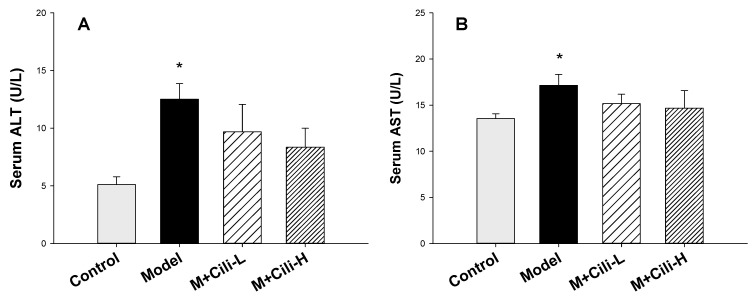
Serum alanine aminotransferase (ALT, (**A**)) and aspartate aminotransferase (AST, (**B**)) determination. Data are mean ± SEM (*n* = 9–11), * Significantly different from the Control group, *p* < 0.05.

**Figure 3 nutrients-14-01974-f003:**
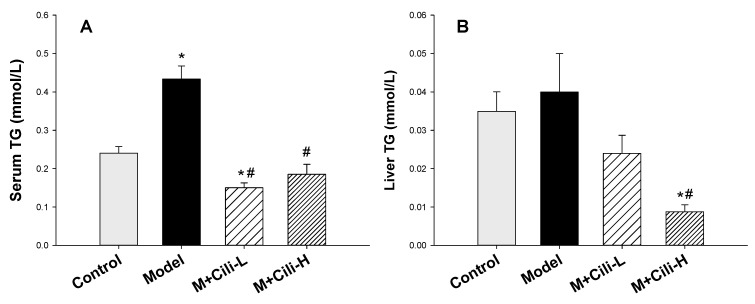
Serum triglyceride (Serum TG, (**A**)) and liver triglyceride (Liver TG, (**B**)) determination. Data are mean ± SEM (*n* = 9–11), * Significantly different from the Control group, *p* < 0.05; # Significantly different from the Model group, *p* < 0.05.

**Figure 4 nutrients-14-01974-f004:**
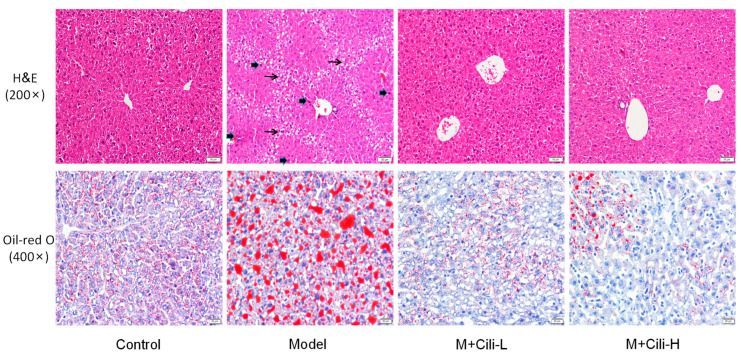
Representative photos of haematoxylin and eosin (H&E) staining (**top**) and Oil-red O staining (**bottom**). Model mice were fed Lieber-DeCarli liquid diet containing 5% alcohol with or without Cili (4 mL and 8 mL/kg for 28 days). Control mice received liquid diet without alcohol. Magnification for H&E (200×), for Oil-red O (400×). Thin arrows indicate hepatocyte vacuolation, slight swelling and degeneration, and thick arrows indicate foci of apoptosis/necrosis.

**Figure 5 nutrients-14-01974-f005:**
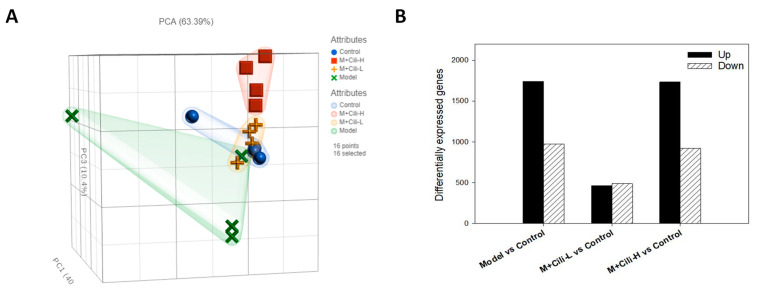
Differentially expressed gene (DEG) analysis. (**A**). Principle Component Analysis (PCA) of 16 samples (4/group); (**B**). Overview of DEGs as compared to the Control group under *p* value ≤ 0.05. Upregulated genes are in black bar, while downregulated genes are in shaded bar.

**Figure 6 nutrients-14-01974-f006:**
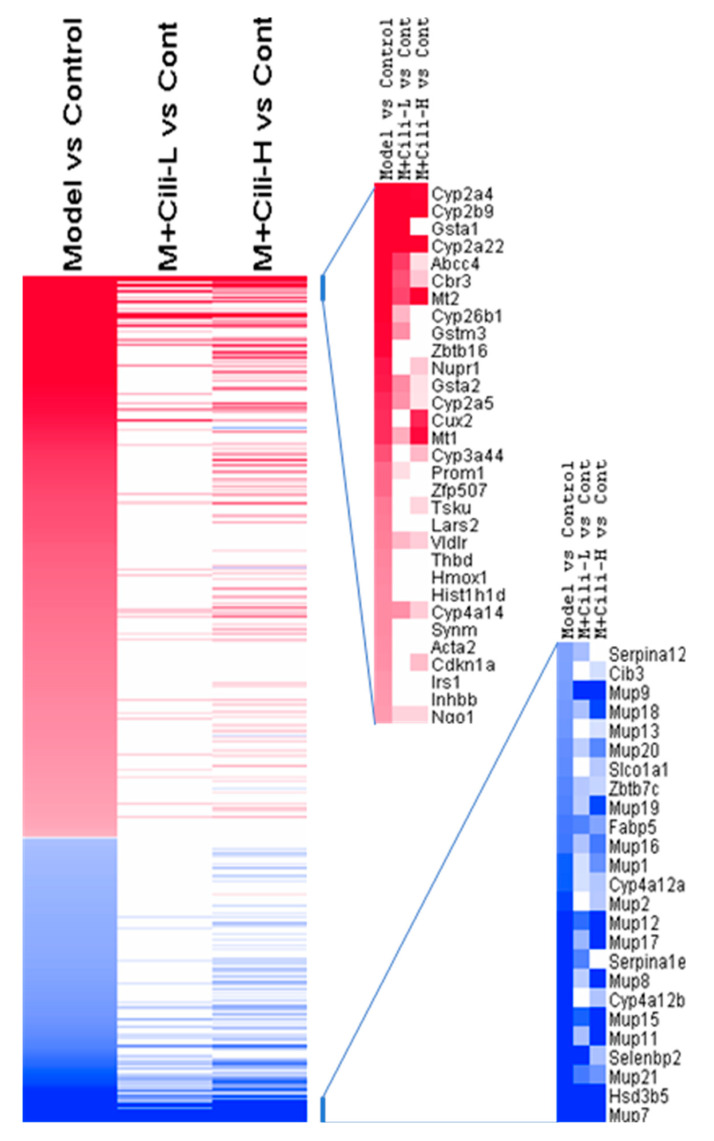
Heatmap comparison of DEGs based on Model vs Control groups. Red indicates increased expression and blue indicates decreased expression. The scale bar was +3/−3 Log2 Fold change. The entire DEGs are provided as Appendix A.

**Figure 7 nutrients-14-01974-f007:**
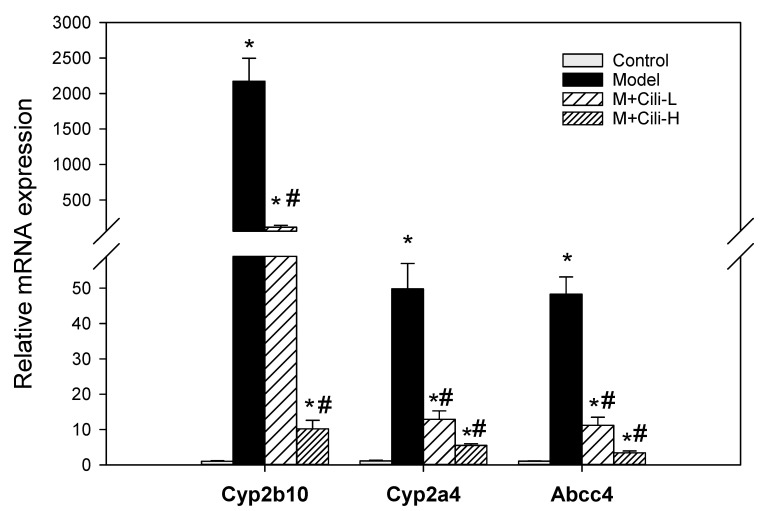
The qPCR analysis of the constitutive androstane receptor (CAR) and pregnane X receptor (PXR) biomarkers among groups. Data are mean ± SEM (*n* = 8), * Significantly different from the Control group, *p* < 0.05; # Significantly different from the Model group, *p* < 0.05.

**Figure 8 nutrients-14-01974-f008:**
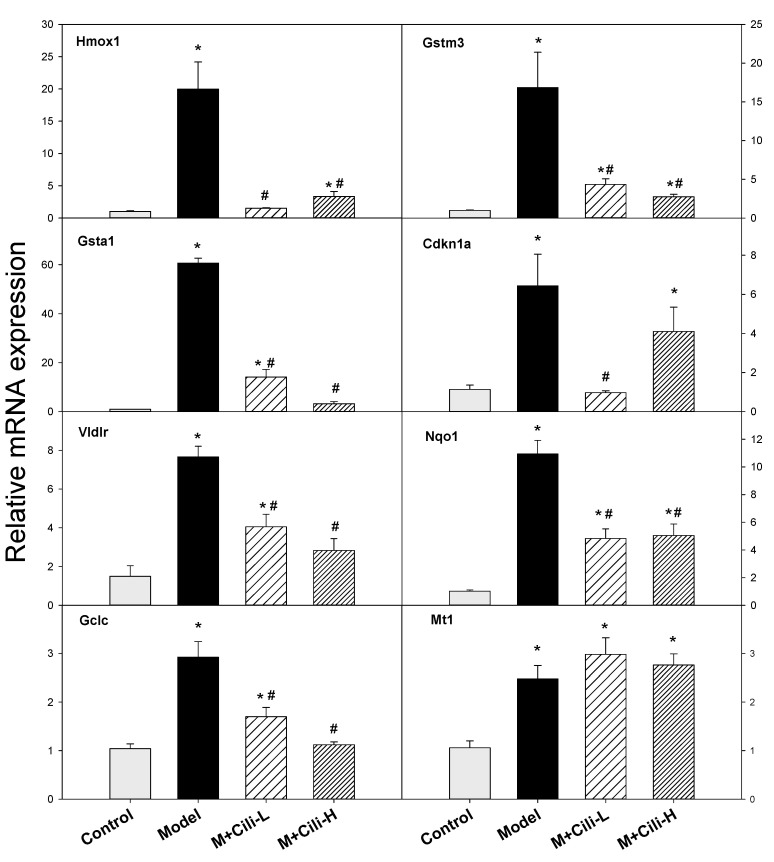
qPCR analysis gene expression related to oxidative damage among groups. Data are mean ± SEM (*n* = 8), * Significantly different from the Control group, *p* < 0.05; # Significantly different from the Model group, *p* < 0.05.

**Figure 9 nutrients-14-01974-f009:**
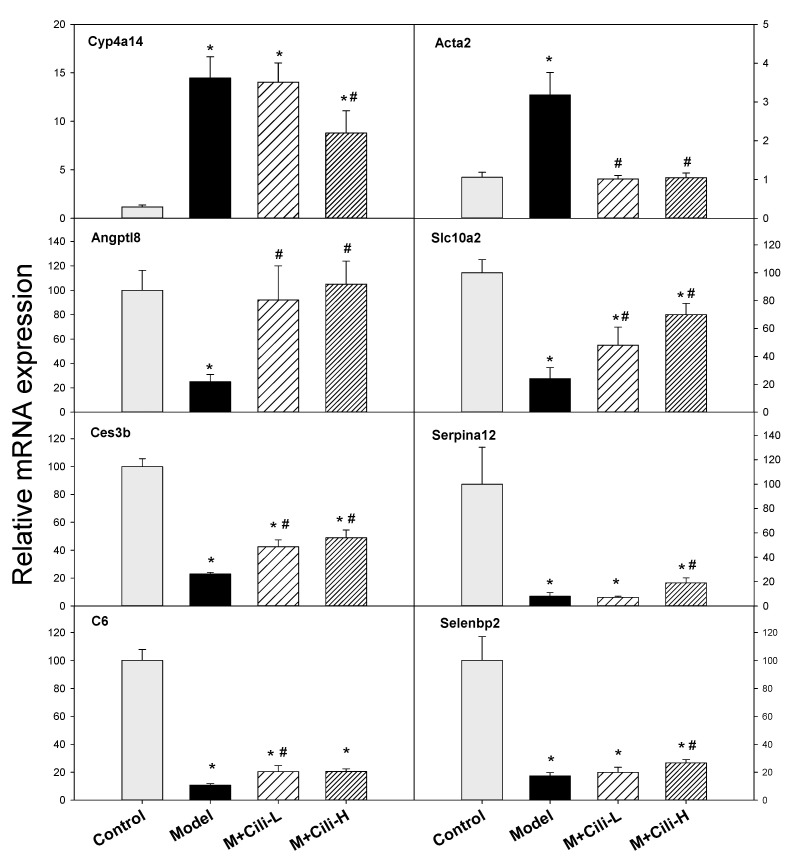
qPCR analysis gene expression related to lipid metabolism related genes among groups. Data are mean ± SEM (*n* = 8), * Significantly different from the Control group, *p* < 0.05; # Significantly different from the Model group, *p* < 0.05.

**Figure 10 nutrients-14-01974-f010:**
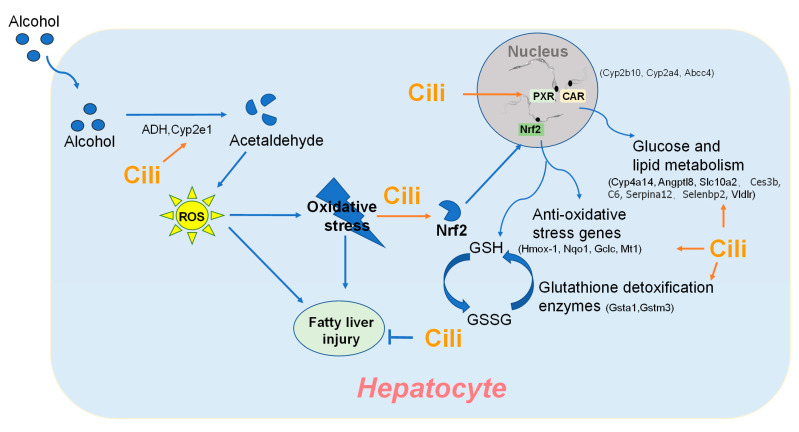
Proposed protective mechanism of Cili against alcoholic liver injury.

## Data Availability

Data are available upon request.

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
