# Peer review of "RNA-Seq Analysis of Protection against Chronic Alcohol Liver Injury by *Rosa roxburghii* Fruit Juice (Cili) in Mice"

_nutrients, 2022, doi:10.3390/nu14091974_

Round 1
Reviewer 1 Report
Manuscript by Yang S et al. “RNA-Seq analysis of protection against chronic alcohol liver injury by Rosa roxburghii fruit juice (Cili) in mice” gives an interesting insight into the effect of Cilli extract in mouse models of alcohol-induced diet.
The improved version did not introduce answers to major issues :.
Major issue:
- QPCR analysis by Syber gree4n methodology – How did you confirm the specificity of your reactions? Please describe . How did you assure that your referent gene GAPDH is stable during treatments?
- All comparisons are performed related to the control group that indicates global effects. To filtrate genes affected by Cili treatment provide the additional figure with the direct effect of Cilli –treatment by comparing the following groups: “M-Cili –L” vs Model and “M-Cili-H” vs Model.
- Are the same genes affected by L and H dose ? … Introduce these results in discussion.
Author Response
Reviewer 1
- QPCR analysis by Syber gree4n methodology – How did you confirm the specificity of your reactions? Please describe. How did you assure that your referent gene GAPDH is stable during treatments?
qPCR is now a routine technique for gene expression studies. For each gene amplification, the dissociation curve and reaction peak were generated to confirm the specificity. A specific primer was designed for each specific gene, and the melting curve and a specific peak are used to confirm the specificity of reaction (see examples below. If the primer is non-specific there will be more than one peak). qPCR uses threshold CT values to evaluate the gene amplification capacity using the 2−△△CT method. Gapdh is a widely used housekeeping gene which is expressed at high levels in almost all tissues, and is not affected by the pathology and pharmacological means. Attached are the Gapdh Ct values of 32 samples from 4 groups with mean ± SD of 19.79 ± 0.63, very stable (See Figure below).
In the revised manuscript we have added: “The specificity of the each gene expression was confirmed by the melting curve with a single peak, and the stability of Gapdh of 32 samples from 4 groups had mean ± SD of 19.79 ± 0.63”.
- All comparisons are performed related to the control group that indicates global effects. To filtrate genes affected by Cili treatment provide the additional figure with the direct effect of Cilli –treatment by comparing the following groups: “M-Cili –L” vs Model and “M-Cili-H” vs Model.
Thanks for the comments. However, no clear dose-response was evident for majority of parameters measured in the present study. Therefore, we have made only two major comparisons: *Significantly different from Control, p < 0.05; #Significantly different from Model, p < 0.05. We think these comparisons are succinct and sufficient.
- Are the same genes affected by L and H dose ? … Introduce these results in discussion.
We have discussed this issue in the Results and Discussion sections.
“Generally speaking, Cili at both doses were effective in ameliorating alcohol-increased liver index, ALT, and triglyceride, but no dose-response was evident”
“Both Cili doses attenuated their over-expression, although no clear dose-response, implying a reduced oxidative damage to the liver”.
Reviewer 2 Report
The article is of interest and of potential translational relevance. Please find below some points that require further action to make the work more scientifically sound.
The text should all be in black and not with parts highlighted in red.
Reference 4 should report the surname of the first author and not the journal name.
How was the sample size determined?
2.8. Heatmap visualization of differentially expressed genes.
P<0.05 seems like a broad threshold with a very high false positive. Data should be corrected for multiple testing (i.e Bonferroni, FDR).
3.4. qPCR analysis of selected genes.
Please explain better in the text why you focused on these genes.
Author Response
Reviewer 2
The text should all be in black and not with parts highlighted in red.
“Red” is a highlighted copy. We have tried to avoid the confusion of a highlighted copy and a clean copy by uploading the highlighted copy as a Supplementary file.
Reference 4 should report the surname of the first author and not the journal name.
Corrected as “by Wang et al., [4]”.
How was the sample size determined?
The sample size in experiment should be ≧6. According to the uncertainty in the experimental process, our proposed sample size is 10-12 samples, and random sampling and random grouping are adopted.
2.8. Heatmap visualization of differentially expressed genes. P<0.05 seems like a broad threshold with a very high false positive. Data should be corrected for multiple testing (i.e Bonferroni, FDR).
Thanks for your comments. The RNA-Seq data is now re-analyzed using FDR (Padj) < 0.05, significantly reducing DEGs. The Fig. 5B, Fig. 6, Supplementary Table 1 and Result description are now substantially revised.
3.4. qPCR analysis of selected genes. Please explain better in the text why you focused on these genes.
In the revised manuscript, the rationale for selected genes are explained, based on (1) the most dramatic changes DEGs (Fig. 6 and Supplementary Table 1), and (2) known biomarkers for alcoholic liver injury. For example in the result description of Figure 7: “Based on the fold change and alcohol liver injury associated with nuclear receptor biomarkers”
Reviewer 3 Report
The manuscript describes the effects of Rosa roxburghii fruit juice on the pathological and transcriptomic phenotypes of a murine model of alcoholic liver disease. There are multiple problems in the paper as listed below.
Specific issues:
1. The dose of pentobarbital for the euthanasia of mice (65 mg/kg) is considered very low. The recommended dose for the euthanasia in most universities of western countries (searchable online) is more than 3 times (≥150 mg/kg) the anesthetic dose (50 mg/kg). The 65 mg/kg is not an overdose. I am concerned that the euthanasia was done inappropriately. How was the dose determined? Was the dose determination really in accordance with the Chinese guidelines for animal welfare? Was the dose approved by the Institutional Animal Care and Use Committee (IACUC)?
2. Data of absolute liver weight should also be added to Fig. 1 as relative liver weight (Fig. 1B) was biased by decreased body weight. Both data are needed to investigate liver pathology.
3. I could not see clear evidence of hepatocellular swelling, necrosis, or inflammatory infiltrates in the liver of Model group (Fig. 4) (the reviewer is a board-certified veterinary pathologist with expertise in rodent models of fatty liver). I just see some cytoplasmic vacuoles in hepatocytes (probably in zone 2?).
4. Additionally, the degree of liver injury was mild in this model as the serum transaminase levels were only < 3 times and < 1.5 times the levels of ALT and AST, respectively. The liver disease in this model would be insufficient. I guess that longer term experiment is needed to induce sufficient degree of alcoholic liver disease.
Author Response
Reviewer 3
- The dose of pentobarbital for the euthanasia of mice (65 mg/kg) is considered very low. The recommended dose for the euthanasia in most universities of western countries (searchable online) is more than 3 times (≥150 mg/kg) the anesthetic dose (50 mg/kg). The 65 mg/kg is not an overdose. I am concerned that the euthanasia was done inappropriately. How was the dose determined? Was the dose determination really in accordance with the Chinese guidelines for animal welfare? Was the dose approved by the Institutional Animal Care and Use Committee (IACUC)?
Thanks for the comments. The reviewer is correct that in most universities of western countries the doses of pentobarbital injection for euthanasia are over 150 mg/kg (Laferriere and Pang, 2020, PMID 32156325), and currently not recommend to use for euthanasia anymore, including in Zunyi Medical University. However, when we conducted the study two years ago, no such regulation was placed in China (see the following two literatures in Chinese with DOI numbers). In complying with the Animal Welfare policy, the use of pentobarbital was removed as “mice were euthanized and blood was collected”.
[1]余婷,张妮,陈思羽,王笑笑,左思洋,郭兵,刘丽荣.猫爪草对缺血再灌注小鼠急性肾损伤的作用及机制[J].贵州医科大学学报,2021,46(11):1301-1308.DOI:10.19367/j.cnki.2096-8388.2021.11.010.
[2]辛悦,于博文,郭鱼波,刘宇飞,裴晓华,樊英怡.舒肝颗粒对肝郁气滞型乳腺增生症小鼠的治疗作用及机制研究[J].北京中医药,2021,40(10):1077-1082.DOI:10.16025/j.1674-1307.2021.10.007.
- Data of absolute liver weight should also be added to Fig. 1 as relative liver weight (Fig. 1B) was biased by decreased body weight. Both data are needed to investigate liver pathology.
Absolute liver weight is affected by body weight, while the liver index is the ratio of liver weight to body weight, which is a more visible and meaningful indicator. The data of the body weights and liver weights at the end of experiment are now added to the Result description.
- I could not see clear evidence of hepatocellular swelling, necrosis, or inflammatory infiltrates in the liver of Model group (Fig. 4) (the reviewer is a board-certified veterinary pathologist with expertise in rodent models of fatty liver). I just see some cytoplasmic vacuoles in hepatocytes (probably in zone 2?).
Thanks for the comments. The reviewer is correct that hepatocytes vacuoles were the major pathology lesion, and only foci of apoptosis/necrosis could be seen. In the revised manuscript, we have used arrows to indicate such lesions in the Figure legends “Thin arrows indicate hepatocyte vacuolation, slight swelling and degeneration, and thick arrows indicate foci of apoptosis/necrosis”. We have softened the result description as “Model group had fat vacuoles, liver degeneration and foci of apoptosis/necrosis compared with Control”.
- Additionally, the degree of liver injury was mild in this model as the serum transaminase levels were only < 3 times and < 1.5 times the levels of ALT and AST, respectively. The liver disease in this model would be insufficient. I guess that longer term experiment is needed to induce sufficient degree of alcoholic liver disease.
Thanks for the comments. We have followed the protocols of the typical 4-week alcohol exposure to produce fatty livers, but not for severe liver injury, fibrosis and cirrhosis. Lieber-DeCarli liquid diet-induced alcoholic fatty liver is characterized by fat droplet accumulation, not by ALT elevation, as only foci of necrosis can be observed [Ref 15-17]. Importantly, mild liver injury model is ideal to evaluate nutritional intervention, as sever liver injury cannot be prevented or cured by many nutrients like Cili juice.
Reviewer 4 Report
The topic of the original research manuscript, which deals with hepatoprotective effects of the plant Rosa roxburghii, is interesting. Since the liver diseases lead often to total metabolic impairments of organism, the topic of this paper is timely and important.
The authors used adequate methods, which were well-established. Additionally, the results were supported by experimental data and properly discussed. The authors cited relevant references. Furthermore, the results were presented by many illustrative figures.
Author Response
Reviewer 4
The topic of the original research manuscript, which deals with hepatoprotective effects of the plant Rosa roxburghii, is interesting. Since the liver diseases lead often to total metabolic impairments of organism, the topic of this paper is timely and important.
The authors used adequate methods, which were well-established. Additionally, the results were supported by experimental data and properly discussed. The authors cited relevant references. Furthermore, the results were presented by many illustrative figures.
Thank you very much for positive remarks.
Round 2
Reviewer 3 Report
Thank you for the responses to the suggestions. I have largely agreed with the revision; however there are still some problems to be solved.
- I do not think that the hepatocytes indicated by thick arrows have apoptosis/necrosis. Apoptosis and necrosis are morphologically distinct. And I could not see clear apoptosis or necrosis in Fig. 4. See the attached PDF.
Author Response
I do not think that the hepatocytes indicated by thick arrows have apoptosis/necrosis. Apoptosis and necrosis are morphologically distinct. And I could not see clear apoptosis or necrosis in Fig. 4. See the attached PDF.
Thanks for the comments with TYPICAL apoptosis and necrosis morphology in PDF.
Apoptosis can be seen by light microscopy, sometimes with the aid of specific dyes (TUNEL), and some changes can be detected only by electron microscopy. The cell dying by apoptosis occurs at various stages. Early changes involve condensation of nuclear chromatin along the perimeter of the nucleus and, dependent on the context, separation from the surrounding cells in the tissue and from extracellular matrix attachments. Blebs appear, and the nucleus condenses completely and segregates into several fragments. Organelles are generally intact although dilatation of the ER and release and aggregation of ribosomes have been observed. Sometimes cytoplasmic vacuoles can be seen. The cell disintegrates into apoptotic bodies which can contain any part of the cellular material. In the final step, apoptotic bodies are taken up by other cells and digested via a lysosomal pathway
Necrosis is an uncontrolled cell death that results in swelling of the cell organelles, plasma membrane rupture and eventual lysis of the cell, and spillage of intracellular contents into the surrounding tissue leading to tissue damage. Coagulative necrosis appear anucleate, eosinophilic, with preserved structure. In liquefactive necrosis the dying cells are digested by hydrolytic enzymes and hence lose their structural integrity and turn into a viscous mass and the accumulation of such necrotic material.
In many cases, apoptosis and necrosis coexist. For the thick labels in Fig. 4, We have consulted Dr. Ling Wang, Chair of Pathology of Zunyi Medical University; Dr. Shaojun Liu, Director of New York City Health hospitals ; and Dr. Sheng Song, University of North Carolina Hospital, all three agreed that these thick labels indicate apoptosis/necrosis. Some even pointed out even typical apoptosis/necrosis (above slide). Thus, we would like to keep Fig. 4 as it was.
This manuscript is a resubmission of an earlier submission. The following is a list of the peer review reports and author responses from that submission.
Round 1
Reviewer 1 Report
This study by Yang et al, demonstrates the protective effect of Cili fruit juice against alcohol-induced liver injury in mice. The authors demonstrate improved triglyceride levels in Cili treated mice. RNA-seq and q-PCR also demonstrated that Cili attenuated many alcohol-induced changes in gene expression. The article is largely clear, but I have the following concerns
Major Issues
- In the discussion, line 297-298 states “clearly demonstrated the protective effects of Cili against chronic 297 alcohol liver injury, evidenced by serum enzyme activities”. However, figure 1 shows no significant difference in either AST or ALT between Model and Cili treated groups. You do not have strong evidence that Cili has protected against liver injury, as measured by transaminases. This claim is repeated in line 320. Make sure that you are not overstating your claims.
- Figure 4 – text in line 214 states that both high and low dose Cili were equally effective at reducing lipid droplet accumulation but the M+Cili-H image from Oil-red O staining shows much greater lipid accumulation than M+Cili-L image. Please ensure that your chosen images are representative of the results
- The introduction, in particular, would benefit from thorough proof-reading for English. There are a number of minor errors such as the inconsistent use of tenses, capitalisations, missing words, occasional awkward paragraph construction etc that detracts from the readability of the introduction. In saying this, the standard of English in the rest of the manuscript is very good.
- Consider including a figure demonstrating the hypothesised mechanism by which Cili has its protective effects
Minor issues
- Line 77 – consider rephrasing “In addtion to routine protective measures” to something like “In addition to routine mesaures of injury”
- Line 132 should say OCT not OTC
- Line 177 should say Tukey not Turkey
- Figure 4 image is not high enough resolution and the labelling is blurry
- Figure 5 - x-axis labels should say Cili not chilli
- Line 367 should say vinegar not vineger
- Line 110 – please state method of euthanasia and route of blood collection
Author Response
Reviewer 1
This study by Yang et al, demonstrates the protective effect of Cili fruit juice against alcohol-induced liver injury in mice. The authors demonstrate improved triglyceride levels in Cili treated mice. RNA-seq and q-PCR also demonstrated that Cili attenuated many alcohol-induced changes in gene expression. The article is largely clear, but I have the following concerns
Thank you for general remarks on this study
- In the discussion, line 297-298 states “clearly demonstrated the protective effects of Cili against chronic 297 alcohol liver injury, evidenced by serum enzyme activities”. However, figure 1 shows no significant difference in either AST or ALT between Model and Cili treated groups. You do not have strong evidence that Cili has protected against liver injury, as measured by transaminases. This claim is repeated in line 320. Make sure that you are not overstating your claims.
The Liebber-DeCaeli alcohol diet-induced chronic alcoholic liver injury involves mild liver damage characterized by steatosis and mild inflammation, which is quite different from acute liver injury induced by hepatotoxicants (i.e., CCl4, acetaminophen). In this case, serum enzyme activities are not a sensitive marker for the liver injury. Serum and liver triglyceride levels, and histopathology (H&E) and fatty live (Oil-red O staining) are more important parameters. In the revised manuscript, we have soften the sentence in the first paragraph of Discussion as “The present study clearly demonstrated the protective effects of Cili against chronic alcohol liver injury, as evidenced by reduced serum and liver triglyceride levels, and importantly by H&E and Oil-red O staining”. And in the Conclusion “demonstrate the protective effects of Cili against chronic alcohol liver injury by routine measures, especially by serum and liver triglyceride levels, histopathology and Oil-red O staining.
- Figure 4 – text in line 214 states that both high and low dose Cili were equally effective at reducing lipid droplet accumulation but the M+Cili-H image from Oil-red O staining shows much greater lipid accumulation than M+Cili-L image. Please ensure that your chosen images are representative of the results
In Figure 4, lipid droplet accumulation was markedly ameliorated following Cili treatment, and there was no apparent difference between the low and high dose groups. The images are representatives of the results, and we would like to keep them as they were.
- The introduction, in particular, would benefit from thorough proof-reading for English. There are a number of minor errors such as the inconsistent use of tenses, capitalisations, missing words, occasional awkward paragraph construction etc that detracts from the readability of the introduction. In saying this, the standard of English in the rest of the manuscript is very good.
We have made careful proof-reading on the entire manuscript to minimize English errors.
Consider including a figure demonstrating the hypothesised mechanism by which Cili has its protective effects
Thanks for excellent suggestion and a summary figure is now included as Fig. 10.
Minor issues
- Line 77 – consider rephrasing “In addtion to routine protective measures” to something like “In addition to routine mesaures of injury”
Corrected.
- Line 132 should say OCT not OTC
Corrected.
- Line 177 should say Tukey not Turkey
Corrected.
- Figure 4 image is not high enough resolution and the labelling is blurry
The high resolution Figure 4 is now loaded separately.
- Figure 5 - x-axis labels should say Cili not chilli
Figure 5 Labels were corrected.
- Line 367 should say vinegar not vinegar
Corrected
- Line 110 – please state method of euthanasia and route of blood collection
Corrected as “mice were euthanized by overdose of pentobarbital (65 mg/kg, ip) and blood was collected via orbital venous plexus. Liver was isolated, weighed, and stored in -80 °C for RNA analysis”.
Reviewer 2 Report
Manuscript by Yang S et al. “RNA-Seq analysis of protection against chronic alcohol liver injury by Rosa roxburghii fruit juice (Cili) in mice” gives an interesting insight into the effect of Cilli extract in mouse models of alcohol-induced diet.
The manuscript is well written but needs some improvement.
Major issue:
- QPCR analysis by Syber gree4n methodology – How did you confirm the specificity of your reactions? Tm and Polyacrilamide gel pictures demonstrating only one band are needed.
- All comparisons are performed related to the control group that indicates global effects. To filtrate genes affected by Cili treatment provide the additional figure with the direct effect of Cilli –treatment by comparing following groups: “M-Cili –L” vs Model and “M-Cili-H” vs Model.
- Are the same genes affected by L and H dose ? … Introduce these results in discussion.
Minor issues:
1. Typing errors ex-line 63 Alcoholism.-
2. Please define “ line 187” what is the liver index ? Fig1B shows liver/body ratio.
Author Response
Reviewer 2
Manuscript by Yang S et al. “RNA-Seq analysis of protection against chronic alcohol liver injury by Rosa roxburghii fruit juice (Cili) in mice” gives an interesting insight into the effect of Cilli extract in mouse models of alcohol-induced diet.
The manuscript is well written but needs some improvement.
Thank you for the positive comments
- QPCR analysis by Syber gree4n methodology – How did you confirm the specificity of your reactions? Tm and Polyacrilamide gel pictures demonstrating only one band are needed.
qPCR is now a routine technique for gene expression studies. For each gene amplification, the dissociation curve and reaction peak were generated to confirm the specificity. See an example below:
- All comparisons are performed related to the control group that indicates global effects. To filtrate genes affected by Cili treatment provide the additional figure with the direct effect of Cilli –treatment by comparing following groups: “M-Cili –L” vs Model and “M-Cili-H” vs Model.
In this study, we have made two major comparisons: *Significantly different from Control, p < 0.05; #Significantly different from Model, p < 0.05. We do not think it is necessary to compare low-dose and high-dose groups, as in many cases, such differences were not evident.
- Are the same genes affected by L and H dose ? … Introduce these results in discussion.
In many cases, such differences were not evident. We have added the statement in the Results and Discussion.
Minor issues:
- Typing errors ex-line 63 Alcoholism.
Corrected
- Please define “ line 187” what is the liver index ? Fig1B shows liver/body ratio
Corrected as “Fig. 1B shows the liver index (Liver/body weight, mg/g)”
Reviewer 3 Report
Summary
The authors made an effort to investigate yet another plant based extract on its effect on liver pathology of mice on an alcohol containing diet versus controls.
The damages induced by alcohol intake in mice are well described and mostly reflect pathophysiology of fatty liver disease, i.e. increased neutral lipid accumulation in the liver, liver cellular damage (as indicated by transaminase elevation. The Cili extract (applied by oral gavage) ameliorated few of the markers such as serum and liver triglyceride content as well as hepatocyte damage (both dosages). Body weight remained unchanged. The study is complemented by RNA seq data from primary liver tissue. PCA analysis allowed separation of the groups based on treatment and a number of genes were deregulated under alcohol intake and improved by Cili treatment.
The study tests the effect of yet another plant based compound on alcoholic liver disease. The study is interesting but stops basically at the point the story is preliminary and immature.
- A control group (standard diet) of course must also be treated with the Cili compound
- The dose of the extract seems to be chosen arbitrarily. It needs to be test at what dose an effect could be produced and when the compound becomes toxic.
- Statistical tests on triglycerides must be re-done and performed using non-parametric methods. Alternatively, all TG levels must be transformed into Log values. Triglycerides are not normally distributed.
- the RNS seq data are ice but from homogenates. There is no cell specificity whatsoever. There is however, published standard data (https://gtexportal.org/home/) where the obtained gene list can be run against cell specific RNA expression to filter for cell specific effects.
- The in vivo study must then be complemented by in vitro studies on hepatocytes to test the effects on common mechanisms involved in lipid accumulation in liver cells and tot test for specific anit-oxidative mechanisms.
Author Response
Reviewer 3
The authors made an effort to investigate yet another plant based extract on its effect on liver pathology of mice on an alcohol containing diet versus controls.
The damages induced by alcohol intake in mice are well described and mostly reflect pathophysiology of fatty liver disease, i.e. increased neutral lipid accumulation in the liver, liver cellular damage (as indicated by transaminase elevation. The Cili extract (applied by oral gavage) ameliorated few of the markers such as serum and liver triglyceride content as well as hepatocyte damage (both dosages). Body weight remained unchanged. The study is complemented by RNA seq data from primary liver tissue. PCA analysis allowed separation of the groups based on treatment and a number of genes were deregulated under alcohol intake and improved by Cili treatment.
The study tests the effect of yet another plant based compound on alcoholic liver disease. The study is interesting but stops basically at the point the story is preliminary and immature.
Thanks for general comments on this study.
- A control group (standard diet) of course must also be treated with the Cili compound
Thanks for the suggestion. However, the major focus of the study is to examine the effects of Cili on alcoholic liver injury, similar to the studies of Cili on High-Fat-Diet-induced hyperlipidemia (Song et al., 2020, 2021, Ref 6 and 7), and against arsenic-induced liver injury (Xu et al., 2021, Ref 11). Nonetheless, we will take this suggestion in future studies.
- The dose of the extract seems to be chosen arbitrarily. It needs to be test at what dose an effect could be produced and when the compound becomes toxic.
We have added in Methods that “The dose of Cili was based on the literature [6,7,11] and our preliminary studies against alcoholic liver injury”.
- Statistical tests on triglycerides must be re-done and performed using non-parametric methods. Alternatively, all TG levels must be transformed into Log values. Triglycerides are not normally distributed.
TG determination was performed with triglyceride kit from Nanjing Jiancheng Bioengineering Institute, following the instruction of the manufacture. From the literature, kit instructions, and our experience, TG determination is straightforward and does not require specific statistical methods.
- the RNS seq data are ice but from homogenates. There is no cell specificity whatsoever. There is however, published standard data (https://gtexportal.org/home/) where the obtained gene list can be run against cell specific RNA expression to filter for cell specific effects.
RNA-Seq used total RNA from the liver, not from homogenates or specific cells.
- The in vivo study must then be complemented by in vitro studies on hepatocytes to test the effects on common mechanisms involved in lipid accumulation in liver cells and tot test for specific anit-oxidative mechanisms.
We disagree with this comment. The in vivo study weighs more heavily than in vitro studies. RNA-Seq is the way to dissect molecular mechanisms, rather than using in vitro cells. See our recent publications with RNA-Seq technology (Zhang et al., Biomed Pharmacother 2021, 137, 111307), and many other publications.
Round 2
Reviewer 3 Report
The manuscript has minimally been edited. The reply the authors sent to the reviewer´s comments is insufficient, neglecting and is partly clueless. This reviewer certainly does not change the initial recommendation.
In more detail:
Ad 1: The authors believe that picking data from a control group published in other papers may be enough and will respect control groups in their future studies. I.e. they agree with the flaw but still insist in publication?
Ad 2: The reviewer was asking for different dosages in the experiments. The authors have apparently performed “our preliminary studies against alcoholic liver injury”. Ok, but why is that not part of the MS?
Ad 3: This is remarkable: the authors apparently do not know that triglycerides are not normally distributed (while cholesterol for instance is). A non-parametric test is no a “specific statistical methods”. The appropriate methodology could be looked up in text books on biostatistics.
Ad 4: The authors obviously have not understood the question. The homogenates are far from cell-specific. That is why the yield of differentially regulated genes should be run against know cell specific gene expression data (from gtexportal) to dissect the cell-specific effects. This reviewer would like to point out that in liver, there are immune cells, endothelial cells, hepatocytes etc. just for info…..
Ad 5: Why should the reviewer check “many other publications”? The experiment asked to obtain a more cell-type specific insight is a simple task. The phrase “RNA-Seq is the way to dissect molecular mechanisms, rather than using in vitro cells” does not make any sense at all.